# Cyclic Hexapeptide from *Bouvardia ternifolia* (Cav.) Schltdl. and Neuroprotective Effects of Root Extracts

**DOI:** 10.3390/plants12142600

**Published:** 2023-07-10

**Authors:** Yury Maritza Zapata Lopera, Gabriela Trejo-Tapia, Manasés González-Cortazar, Maribel Herrera-Ruiz, Alejandro Zamilpa, Enrique Jiménez-Ferrer

**Affiliations:** 1Centro de Investigación Biomédica del Sur, Instituto Mexicano del Seguro Social, Xochitepec 62790, Morelos, Mexico; yzapatal1600@alumno.ipn.mx (Y.M.Z.L.); gmanases@hotmail.com (M.G.-C.); cibis_herj@yahoo.com.mx (M.H.-R.); azamilpa_2000@yahoo.com.mx (A.Z.); 2Centro de Desarrollo de Productos Bióticos, Instituto Politécnico Nacional, Yautepec 62730, Morelos, Mexico

**Keywords:** GC–MS, NMR, neuroinflammation, blood-brain barrier, lipopolysaccharide

## Abstract

*Bouvardia ternifolia* (Cav.) Schltdl. is a shrub that belongs to the *Rubiaceae* family and is distributed throughout México; it has been used for its antioxidant, neuroprotective, and anti-inflammatory properties. This work aimed to evaluate the protective effects of *B. ternifolia* root extracts on the blood-brain barrier and the positive regulation of cytokines IL-1β, IL-6, and TNF-α, and the characterization of compounds present in the dichloromethane (BtD) and hexane (BtH) extracts. Male ICR mice were orally administered with *B. ternifolia* extracts for 5 days before a single injection of LPS. Administration of BtH and BtD significantly decreased Evans blue leakage into brain tissue by 70% and 68%, respectively. Meloxicam (MX) decreased the concentration of IL-1β by 39.6%; BtM by 53.9%; BtAq by 48.4%; BtD by 31.9%, and BtH by 37.7%. BtH was the only treatment that significantly decreased the concentration of IL-6 by 32.2%. The concentration of TNF-α declined with each of the treatments. The chemical composition of BtD and BtH was characterized by GC–MS, and the cyclic hexapeptide was identified by ^13^C, ^1^H NMR, and two-dimension techniques. In the BtD extract, seven compounds were found and in BtH 13 compounds were found. The methanolic (BtM) and aqueous (BtAq) extracts were not subjected to chemical analysis, because they did not show a significant difference in the BBB protection activity. Therefore, the results suggested that the extracts BtD and BtH protect the blood-brain barrier, maintaining stable its selective permeability, thereby preventing LPS from entering the brain tissue. Simultaneously, they modulate the production of IL-1β, IL-6, and TNF-α. It is important to note that this research only evaluated the complete extracts.

## 1. Introduction

Neurodegenerative diseases are predicted to be the most significant health concern in this century and the second leading cause of death by 2050. Related to the increase in life expectancy, there has been a surge in the incidence of these disorders. Neurodegenerative diseases encompass a range of conditions in which neuronal structure and function are altered, affecting the brain and spinal cord central nervous system (CNS) and worsening over time [1]. The main cellular and molecular events that cause neurodegeneration are oxidative stress, deposition of protein aggregates, neuroinflammation, impaired mitochondrial function, induction of apoptosis, and alteration of autophagy [2].

Lipopolysaccharides (LPS) are endotoxins formed of an O-antigen and are found in the outer membrane of Gram-negative bacteria and play a role as stimulants for microglial activation. Toll-like receptor 4 (TLR4) is expressed on microglial cells and is responsible for the inflammatory cascade in microglia by binding to LPS [3].

Active microglia initiate a process of brain inflammation and play a crucial role in regulating neuroinflammatory reactions [4]. Hyperactive microglia are known to release a variety of neurotoxic mediators, such as nitric oxide (NO), inducible nitric oxide synthase (iNOS), cyclooxygenase-2 (COX-2), prostaglandin E2 (PGE2), and a series of pro-inflammatory cytokines including tumor necrosis factor α (TNF-α), interleukin-6 (IL-6), and interleukin-1β (IL-1β), which further lead to neurodegenerative disorders [5]. Once LPS binds to TLR4 on the surface of microglia, it activates several signal transduction pathways of inflammatory processes [6]. Activated TLR4 transfers the signal through the two main downstream pathways: first, the TLR4-mediated myeloid differentiation factor 88 (MyD88), dependent pathway, and second, the Toll/IL-1 receptor domain-containing adapter induction of the interferon-β (TRIF)-dependent pathway [7]. MyD88 is an adapter protein that mediates signaling pathways for most TLRs, activating NF-κB and MAPKs [8].

*Bouvardia ternifolia* (Cav). Schldtl. is a shrub with red flowers and abundant roots that receives the Nahuatl name of ‘ezpahtli’ (blood medicine) and is described in the Libellus de medicinalibus indorum herbis, better known as the De la Cruz-Badiano Codex. It is distributed throughout México. In traditional medicine, the infusion of the aerial parts of the plant is used to alleviate various conditions, such as digestive (stomachache, diarrhea, and dysentery), respiratory (cough, whooping cough), and liver problems. It is also used in the treatment of tumors and leukemia. In addition, it is used as a bile tonic, in the control of diabetes, as a sedative for nervous problems, as an organic tonic, in the treatment of pimples or vaginal abscesses, to reduce fever and headaches, as an anti-inflammatory for blows, as an analgesic, and in the form of a poultice against snake, scorpion, and spider bites. In addition, the ancient indigenous people used root powder to heal sores, stop bleeding (hemostatic), heat and heart exhaustion, and scorpion stings [9]. The pharmacological activities of *B. ternifolia* are cytotoxic, antitumor [10], antivenom [11], anti-inflammatory, and NF-kB inhibition [12]. In this last activity, the root of the plant is used, and the competitive inhibition activity of the enzyme acetylcholinesterase has been shown [13]. In addition, the aerial parts of the plant have shown antioxidant and anti-inflammatory activity [14]. The pharmacological properties discussed above are due to specialized metabolites. The presence of metabolites such as bouvardin, scopoletin, ternifolial, and ternifoliol has been reported in the root [12].

Chemical studies carried out on the methanolic extract of the leaves, flowers, and stems of the species have revealed the presence of three cyclic hexapeptides: bouvardin, deoxy-bouvardin, and 6-O-methylbouvardin [10,15]. These hexapeptides comprise three alanine units, two L-alanine and one D-alanine, and three N-methyltyrosine units. These components combine to form an 18-membered ring connected to another 14-membered ring. The linkage between two adjacent tyrosine units involves a phenolic oxygen bond, and a *cis* peptide bond that contributes to the rigidity and stability of the molecular structure. The presence of triterpenic acids, such as ursolic acid and oleanolic acid, has been reported in the hexanic and methanolic extracts of the roots, as well as in the chloroformic extract of the stem [16].

This work aimed to evaluate the protective effects of *B. ternifolia* root extracts on the blood-brain barrier and the positive regulation of cytokines IL-1β, IL-6, and TNF-α, as well as the characterization of compounds present in the dichloromethane (BtD) and hexane (BtH) extracts.

## 2. Results

### 2.1. Protective Effect of B. termifolia Root Extracts on Blood-Brain Barrier

Regarding the effect of the treatments on the acute inflammation caused by LPS in the BBB permeability model with Evans blue (Figure 1), a higher concentration of Evans blue per mg of tissue was observed in the damage control group. The treatment applied to the positive control group meloxicam (MX) decreased the concentration of Evans blue in the brain. Regarding the experimental treatments, it was observed that BtH significantly decreased by 70% the concentration of Evan’s blue per mg of brain tissue, as did the BtD, by 68%. The BtM was reduced by 13%, and the aqueous BtAq by 22%. BtD and BtH had a statistically significant difference with respect to the MX group, meaning that these *B. ternifolia* treatments have a higher percentage of reduction in the extravasation of Evans blue than the treatment with meloxicam.

### 2.2. Effect of B. ternifolia Root Extracts on Cytokine Levels in the Brain of LPS-Administered Mice

Figure 2A,B shows that the administration of LPS (5 mg/kg for 4 h) induced a significant increment of IL-1β and IL-6 on the brain in comparison with the basal control group (* *p* < 0.05). However, these IL-1β values significantly decreased with MX by 39.6%, BtM 53.9%, BtAq 48.4%, BtD 31.9%, and BtH 37.7%. IL-6 was only significantly decreased by the BtH extract by 32.2%.

An increment of TNF-α (Figure 2C) in the brain tissue with LPS was observed compared to the basal control group. At the same time, meloxicam showed a decrease in the concentration of TNF-α by 18%, methanol extract by 34%, aqueous extract by 23.4%, dichloromethane extract by 31%, and hexane extracts by 24.3% with respect to the damage control; these values were statistically different with respect to the damage group.

### 2.3. Chemical Analysis

Chemical analysis of BtD and BtH extracts led to the identification of twenty compounds from the root of *B. ternifolia*: one cyclic hexapeptide, seven fatty acids, one alkane, two triterpenes, two phenanthrene-carboxylic acids, one isoprenoid lipid, one phytosterol, one tocopherol, one monoterpenoid, one tetramethyl, and one indanylidene.

#### 2.3.1. GC-MS Analysis of Dichloromethane Extract

The metabolic profile of the plant extracts involves analysis using the gas chromatography technique coupled with mass spectrometry (GC-MS). Following this approach, seven different compounds were identified in the BtD extract (Table 1). The metabolic profile of the extract of *B. ternifolia* included: alkane (3), isoprenoid lipid (5), tocopherol (6), terpenoid (1, 7), tetramethyl (2), and indanylidene (4) compounds (Appendix A).

#### 2.3.2. GC–MS Analysis of Hexane Extract

The analysis of the BtH by GC-MS allowed for the identification of 13 compounds (Table 2). Triterpene (20), tocopherol (18), fatty acids (8, 9, 10, 11, 12, 13, 14), isoprenoid lipid (17), phenanthrene-carboxylic acids (15 and 16), and phytosterol (19); as depicted in Appendix A.

#### 2.3.3. Isolation of Cyclic Hexapeptide

Fraction BtD4.1 was analyzed using HPLC and TLC techniques, followed by acetylation and open-column chromatography separation. Within this fraction, a cyclic hexapeptide (Figure 3) was obtained. The isolated compound was further analyzed by ^13^C and ^1^H-NMR, as well as two-dimensional analysis.

Cyclic hexapeptide was obtained as a deep yellow precipitate that was soluble in dichloromethane and could form microcrystals. TLC detected a blue fluorescent band under UV light (365 nm) and a gray band under ultraviolet light λ = 254 nm. HPLC showed a 9.6 min peak at 270 nm and an absorption spectrum λ_max_ = 211 and 276 nm characteristic of a cyclopeptide (Appendix A). ESI-MS data indicated a positive ion at m/z 934.59 [M + H]^+^, and its molecular formula was C_46_H_58_N_6_O_15_ (Appendix A).

Cyclic hexapeptide is formed by an 18-membered ring consisting of D-Ala^1^, Ala^2^, N-Me Tyr^3^, and Ala^4^, attached to another 14-membered ring. The above was observed from the chemical changes of carbon and proton and the proton coupling constants (Appendix A). The identification of amino acid residues in the compound primarily relied on HMBC correlations between the N-CH_3_ proton and the N-H proton of an amino acid residue, along with the adjacent carbonyl carbon. In the HMBC spectrum, cross-peaks such as Tyr^3^-NMe/Ala^2^-CO, Tyr^5^-NMe/Ala^4^-CO, and Tyr^6^-NMe/Tyr^5^-CO indicated the correlations between Ala^2^-Tyr^3^ and Ala^4^-Tyr^5^-Tyr^6^ (Table 3). Additionally, HMBC correlations were observed between Ala^1^-NH/Tyr^6^-CO, Ala^2^-NH/Ala^1^-CO, and Ala^4^-NH/Tyr^3^-CO. In conclusion, the HMBC correlations confirmed the presence of an 18-membered ring (Appendix A). Furthermore, based on the ^13^C NMR spectrum, a hydroxyl group was assigned to C-6β, which was supported by the ^1^H-^1^H COSY correlation of H-6β/H-6α (^δ^H 4.48 d, *J* = 6.9 Hz) (Appendix A) and HMBC correlations of H-6β/C-6δa, C-6δb, and Tyr^6^ C=O (Figure 4). To determine the presence of glucose, the correlation was observed using HMBC of the anomeric proton (^δ^H 5.0, d, *J* = 7.8 Hz) with C-6ζ indicating that the sugar group was linked to the ζ-position of Tyr6 (Appendix A). According to these analyses, this compound corresponds to a cyclic hexapeptide identified as (1S,4R,7S,10S,13S,16S,28S)-28-hydroxy-10-[(4-methoxyphenyl) methyl]-4,7,9,13,15,29-hexamethyl-24-[(2S,3R,4S,5S,6R)-3,4,5-trihydroxy-6-(hydroxymethyl)oxan-2-yl]oxy-22-oxa-3,6,9,12,15,29-hexazatetracyclo [14.12.2.2^18,21^.1^23,27^] tritriaconta-18,20,23,25,27 (31), 32-hexaene-2,5,8,11,14,30-hexone, commonly known as Rubiyunnanin H.

## 3. Discussion

*B. ternifolia* has been used in traditional medicine to treat inflammation-related conditions. Its anti-arthritic, anti-inflammatory, and inhibitory effect on the NF-kB signaling pathway was recently demonstrated [12]. Cytokines and interleukins contribute to the amplification of inflammatory pathways involved in neurodegenerative disorders, including Alzheimer’s disease, multiple sclerosis, Parkinson’s disease, and amyotrophic lateral sclerosis. LPS, a neurotoxin, is commonly used as an experimental model to induce systemic inflammatory responses. The present study examined the effect of *B. ternifolia* root extracts on LPS-induced damage to the permeability and stability of the blood-brain barrier and pro-inflammatory cytokine alterations in experimental mice. In addition, the extracts that demonstrated greater BBB neuroprotection activity were chemically characterized. Our results showed a decrease in the extravasation of Evans blue into brain tissue, indicating that the BtD and BtH extracts helped to protect the permeability of the blood-brain barrier, as well as the decrease in proinflammatory cytokines in the brain of mice after LPS-induced damage.

Previous studies reported in the literature have shown that LPS induces inflammatory responses both at the systemic level and at the brain level that include the activation of resident immune cells, such as macrophages and microglial cells. These immune cells release a variety of pro-inflammatory molecules, such as cytokines and interleukins, which promote inflammation in the central nervous system [16]. In the present study, the induction of neuroinflammation in the mice brain was shown by the increase in IL-1β, TNF-α, and IL-6 production.

LPS can activate the NF-κB pathway through interaction with its receptor, TLR-4. Once in the nucleus, NF-κB binds to its binding sequence to activate the relevant promoters and cause the expression of inflammatory cytokines, such as IL-1β, IL-6, COX-2, and TNF-α. These facts indicate that NF-κB plays a critical role in the regulation of inflammation, and that NF-κB inhibition may protect against neuroinflammation and neurodegeneration [17]. In this study, the extracts of the root of *B. ternifolia* at a dose of 25 mg/kg attenuated the production of the interleukins IL-1β, IL-6, and TNF-α; the mechanism of action by which the compounds present in the extracts would be acting is by inhibiting the NF-κB pathway. Numerous natural products have demonstrated their potential in exerting anti-neuroinflammatory effects by employing various mechanisms. These mechanisms include inhibiting the activation of microglia, reducing the release of pro-inflammatory cytokines from activated microglia, as well as inhibiting the activation of NF-κB and p38 MAPK [18].

It has been reported that some natural products can interfere with the intracellular signaling pathways that regulate the production and activity of cytokines. For example, they can modulate the signaling pathway of protein kinases (MAPK) or the signaling pathway of Janus kinases (JAK)/signal transducers and activators of transcription (STAT), which are important in the regulation of the expression of cytokines [19].

To assess the effects of *B ternifolia* extracts on the permeability of the blood-brain barrier after intraperitoneal injection with LPS, we evaluated Evans blue extravasation to the brain; dichloromethane and hexane extracts were the most effective treatments compared to the VEH and meloxicam groups (70% and 68%, respectively). Evans blue is a dye that specifically binds to albumin. Under physiological conditions, the endothelium acts as a barrier, preventing the passage of albumin and thus restricting the movement of Evans blue within the blood vessels. However, under pathological conditions characterized by increased vascular permeability, endothelial cells undergo changes that result in the loss of the tight interconnection between them. Consequently, the endothelium becomes permeable to small proteins such as albumin. This altered state allows extravasation of Evans blue from blood vessels into surrounding tissues, including the brain. The level of vascular permeability can be assessed by simple visualization or by quantitative measurement of the dye incorporated per milligram of tissue [20].

When the permeability of the BBB is compromised, it allows the passage of high concentrations of LPS from the bloodstream into the brain, along with inflammatory cells and mediators. This condition results in a significant worsening of neuroinflammation [21]. Several studies have shown that plant extracts have protective effects on the BBB. These extracts contain bioactive compounds, such as polyphenols, flavonoids, and terpenoids, which have antioxidant and anti-inflammatory properties [22].

The protection of the BBB by plant extracts has been attributed to several beneficial actions. First, these extracts can strengthen the tight junctions between the endothelial cells that make up the BBB. Tight junctions play a crucial role in regulating the flow of molecules and cells from the bloodstream to the brain. By strengthening these junctions, plant extracts can help prevent the infiltration of harmful substances or inflammatory cells into the brain [23,24].

Furthermore, plant extracts can modulate the inflammatory response in the BBB. Chronic inflammation can weaken the barrier and compromise its protective function. Compounds present in plant extracts can inhibit the activation of immune cells, such as macrophages and microglial cells, thus reducing the production of pro-inflammatory cytokines and preventing damage to the BBB [24].

Another mechanism by which plant extracts can protect the BBB is through their antioxidant capacity. These compounds can counteract oxidative stress, which is a major cause of barrier damage. By reducing the production of free radicals and promoting a proper redox balance [25]. In this study, the characterization of the compounds present in the BtD and BtH extracts was also carried out by means of gas chromatography analysis coupled to masses. The dichloromethane extract presented a great chemical diversity in its composition, highlighting the presence of seven compounds, including, for example, the α-tocopherol, which has been reported as a neuroprotector as it is a powerful antioxidant that neutralizes reactive oxygen and nitrogen species [26]; squalene, which has been used for its antioxidant properties [27] and immunomodulatory activity; 2-Nonadecanone has antioxidant properties, which may help to protect cells from oxidative damage caused by free radicals, and anti-inflammatory properties [28]; 3-Carene has anti-inflammatory properties and antioxidant activity [29]; lupeol has neuroprotective effects [30].

In the BtH extract, 13 compounds were found. Among them, for example, was ursolic acid, which has been reported to decrease the level of proinflammatory markers such as COX-2, iNOS, TNF-α, IL-1β, IL-2, and IL-6 in the brain of mice [31]; β-sitosterol has been reported to have an anti-inflammatory effect and act as a modulator of the expression of proinflammatory markers, such as IL-6, iNOS, TNF-α, and COX-2. [17].

In the BtD4.1 fraction of the dichloromethane extract, a glycosylated hexapeptide cyclic-type compound was identified; it is the first time that it has been reported for the root of *B. ternifolia*. However, it has been isolated before under the name of Rubiyunnanins H from the roots of *Rubia yunnanensis* (Franch.) Diels, a plant belonging to the Rubiaceae family, such as *Bouvardia ternifolia*. Previous research has identified bicyclic hexapeptides from Rubiaceae family plants, also known as RAs. The first reported RAs were bouvardin and deoxybouvardin, isolated from the stems, leaves, and flowers of *B. ternifolia*. Since then, an additional twenty-eight RAs have been identified in three Rubia plants: *Rubia cordifolia*, *Rubia akane*, and *Rubia yunnanensis*. Rubiyunnanins H has been reported for its cytotoxic activity in in vitro cultures of cancer cell lines, as well as its ability to inhibit nitric oxide production in the LPS model and IFN-ϒ-induced RAW 264.7 murine macrophages, and to inhibit NF-κB and TNF-α activation [32].

In this study, meloxicam was employed as a positive control, representing a nonsteroidal anti-inflammatory drug (NSAID) commonly used for the management of pain, inflammation, and fever. Meloxicam exerts its effects through selective inhibition of the enzyme cyclooxygenase-2 (COX-2), which plays a pivotal role in prostaglandin synthesis. Prostaglandins serve as critical mediators in the inflammatory cascade, contributing to the sensitization of nerve endings to pain and the amplification of the inflammatory response [33]. By targeting COX-2, meloxicam effectively diminishes the production of proinflammatory prostaglandins, leading to a reduction in the overall inflammatory response. Consequently, this includes a decrease in the generation of pro-inflammatory interleukins, such as interleukin IL-1β, IL-6, and IL-8, which are recognized for their involvement in mediating inflammation and facilitating the transmission of pain [34].

## 4. Materials and Methods

### 4.1. Plant Material

Roots (1.7 kg) of *Bouvardia ternifolia* (Cav.) Schltdl. were collected in Huitzilac, Morelos, Mexico (19°1′48′′ N–99°15′56′′ W). The species was identified by biologists Margarita Avilez and Macrina Fuentes, and a voucher specimen was deposited in the INAH Botanical Garden Herbarium (INAH-MOR-2080).

The plant material was air-dried at room temperature for six days. Once dry, it was ground to achieve a particle size of 1–5 mm using a mill (Pulvex S.A de C.V, Mexico City, Mexico). Plant extracts were obtained through a maceration process, starting with hexane (BtH), followed by dichloromethane (BtD), methanol (BtM), and finally, water (BtAq). Each extract underwent filtration, a process repeated three times for each extract. The solvents from the filtrate were recovered through reduced-pressure distillation using a Büchi 490 rotary evaporator (Büchi, Postfach, Flawil, Switzerland). To achieve complete drying, the extracts were lyophilized and stored at −4 °C. Subsequently, thin-layer chromatography (TLC) and high-performance liquid chromatography (HPLC) were employed to monitor the extracts.

### 4.2. LPS-Induced Acute Brain Inflammation Model in Mice

For this assay, female mice of the ICR strain weighing between 30 and 42 g were used and were divided into 7 groups, with 6 mice in each group, for a total of 84 mice. Animals were provided by the animal center of the Health Research Coordination of the Siglo XXI Medical Center (Mexico City) and were strictly handled according to Mexican regulations (NOM-062-ZOO-1999). The protocol was approved by the local Ethics Committee (R-2020-1702-033). The mice were administered with the extracts at a dose of 25 mg/kg vo, twice a day for three days. (Table 4). After the pretreatment, LPS was intraperitoneally administered at 5 mg/kg and left to act for 4 h.

#### 4.2.1. Quantification of Evans Blue Extravasation

A sterile 0.5% Evans blue solution in PBS was prepared and 200 µL of Evans blue solution was intravenously placed in the lateral tail vein of the mouse. The mouse was then placed in the cage and observed for 30 min. The mice had previously been administered the crude extracts for 5 days, and 4 h before the injection with Evans blue they had been injected with LPS at 5 mg/kg i.p. (Table 4).

Mice were euthanized by a lethal dose of pentobarbital at 100 mg/kg i.p., the ribcage was opened and transcardially perfused through the left ventricle with 100–150 mL of saline. A concomitant small cut was made in the right atrium to remove intravascular blood and tracer, and perfusion continued for 15 min until the atrial fluid was clear. Subsequently, the brain was collected in 1.5 mL tubes and weighed, and then 500 µL of formamide was added to each tube and transferred to a 55 °C water bath for 24–48 h to extract the dye from brain tissue. The supernatants were cooled and centrifuged at 14,000× *g* for 15 min. Albumin-Evans blue concentration was spectrophotometrically quantified at 610 nm and using a standard curve [35]. Protein concentration in the samples was quantified using a modified Bradford method [36].

#### 4.2.2. Quantification of Cytokines

Mice were euthanized with a lethal dose of pentobarbital at 100 mg/kg i.p. The ribcage was opened and transcardial perfusion was conducted through the left ventricle using 100–150 mL of saline solution. Simultaneously, a minor incision was performed in the right atrium to extract intravascular blood and Evans blue, while perfusion was maintained for 15 min. Following this, the brain tissue was homogenized in 1× PBS solution containing 0.1% protease inhibitor (phenyl-methyl-sulfonyl fluoride, PMFS from Merck KGaA, Darmstadt, Germany). The homogenate was then subjected to centrifugation at 3000× *g* for 5 min, and the resulting supernatant was preserved at −80 °C until further analysis. Cytokine concentrations (IL-1β, IL-6, and TNF-α) were quantified using ELISA according to the manufacturer’s instructions (Becton, Dickinson and Co., Franklin Lakes, NJ, USA).

### 4.3. GC-Mass Spectrometry (CG-MS)

BtD and BtH extracts (5 mg) were analyzed by GC-MS. Analysis was performed using an Agilent/HP 6890 gas chromatograph coupled to a quadrupole mass spectrometer (5973 MSD) and fitted with a capillary column (5MS-l 30 m × 0.25 mm, i.d.; 0.25 μm film thickness). The oven temperature was programmed at 40 °C for 1 min and was then increased at 10 °C/min to 280 °C. The inlet temperature was set at 250 °C. The mass spectrometer was operated in positive electron impact mode (EI, 70 eV). Samples were injected in 1 μL volume using helium as a carrier gas (1 mL/min). Detection was performed in selective ion monitoring (SIM) mode, and peaks were identified and quantified using target ions. Compound characterization was based on comparing their mass spectra with the National Institute of Standards and Technology (NIST) library version 1.7a. Relative percentages were determined by integrating the peaks using GC Chem Station software (v C.00.01). The composition was reported as a percentage of the total peak area [37].

### 4.4. Fractionation of the BtD Extract

The BtD extract (5.5 g) was fractionated by successive open-column chromatography using silica gel as the stationary phase (60, F254, Merck KGaA, Darmstadt, Germany) and a gradient system n-hexane: ethyl acetate: methanol with ascending polarity. Fractions of 40 mL were collected and pooled into four subfractions (BtD1, BtD2, BtD3, and BtD4). BtD4 (116 mg) was acetylated and subfractioned into four subfractions (BtD4.1, BtD4.2, BtD4.3, and BtD4.4). Separation was monitored by TLC and HPLC. Aluminum plates coated with silica gel 60 F254 (normal phase, Merck) and silica gel 60 RP-18 F254S (reverse phase, Merck) were used. The plates were analyzed with an ultraviolet light lamp (UVGL-58, 254–365 nm UV) and specific developers. HPLC was performed with equipment consisting of a separation module chromatographic system (Waters 2695) and photodiode array detector (Waters 2996), as well as a 250 × 4 mm Licrosphere^®^ 100 RP-18 Column (5 μm particle size). The mobile phase consisted of gradient water: acetonitrile. Samples were analyzed at 400 μg/mL, with a flow of 0.9 mL/min and a sample injection of 10 μL. The detection of compounds was carried out between 195–600 nm [12].

#### Isolation and Identification of Cyclic Hexapeptide

A fractionation of BtD3 (600 mg) was carried out, obtaining 75 fractions; in TLC, it was observed that most of the compounds presented a blue coloration, and it was possible to isolate the BtD4 subfraction. This fraction was analyzed by HPLC and 4 peaks with similar retention times and UV spectra were observed. Therefore, it was necessary to perform acetylation of 50 mg of the fraction and then carry out another chromatographic separation to isolate the four compounds in BtD4. Subsequently, BtD4.1 was analyzed by NMR (^1^H and ^13^C NMR) and two-dimensional techniques (HMBC, HSQC, and COSY).

### 4.5. Nuclear Magnetic Resonance

BtD4.1 was subjected to structural chemical characterization using mass spectrometry techniques. In addition, proton and carbon nuclear magnetic resonance spectra (^1^H and ^13^C NMR), as well as two-dimensional (2-D) correlated spectroscopy (COSY), heteronuclear single quantum coherence (HSQC), and heteronuclear multiple bond coherence (HMBC); experiments were performed using a Varian INOVA-400 instrument at 400 MHz. BtD4.1 was appropriately diluted in deuterated methanol (CD_3_OD). The software ACD/Labs Predictors, ACD/Processor 12.0, and MestreNova LITE were used for subsequent analysis and interpretation, serving as theoretical simulators to elucidate the structure of the compound.

## 5. Statistical Analysis

Results were statistically analyzed using GraphPad^®^ Prism9 (GraphPad Software Inc., La Jolla, CA, USA) or SPSS 22 (SPSS Inc., Chicago, IL, USA). All data are expressed as the mean ± standard error of the mean. The variables were evaluated by one-way analysis of variance (one-way-ANOVA) and the Dunnett post hoc test. Values with (*) *p* < 0.05 (**) *p* < 0.01, (***) *p* < 0.001, (****) *p* < 0.0001 were considered statistically significant.

## 6. Conclusions

In the present study, we report the anti-inflammatory and neuroprotective properties of *B. ternifolia* root extracts and the compounds contained in the active extracts. This study demonstrates that *B. ternifolia* root extracts inhibit the production of the proinflammatory interleukins IL-1β, IL-6, and TNF-α in an LPS-induced mice model. The inhibitory effect of interleukins was attributed to the suppression of the transcriptional activation of NF-κB through its membrane receptor TLR-4 since NF-κB is one of the key transcription factors responsible for regulating inflammation-related genes (Figure 5). In addition, the *B. ternifolia* root extracts that best protected the blood-brain barrier were BtD and BtH. The main compounds found in the BtD extract were Rubiyunnanins H, terpene-type compounds, α-tocopherol, and squalene; the main compounds found in the BtH extract were fatty acid-type compounds, α-tocopherol, β-sitosterol, and terpene-type compounds. In summary, the inhibition of proinflammatory molecules IL-1β, IL-6, and TNF-α through NF-κB by BtD and BtH extracts from the *B. ternifolia* root, as well as the protection of the blood-brain barrier, would be a possible therapeutic approach for the treatment of neuroinflammation.

## Figures and Tables

**Figure 1 plants-12-02600-f001:**
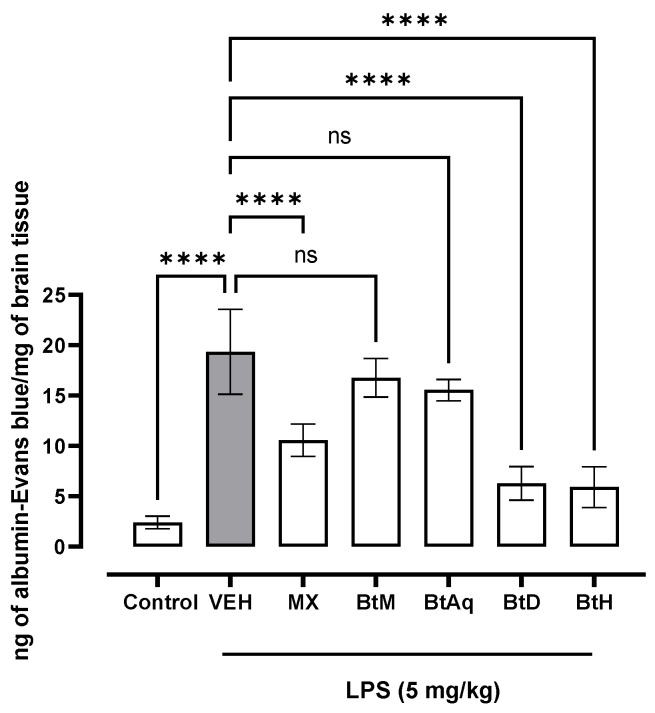
Effect of *B. ternifolia* extracts on Evans blue extravasation to brain tissue with respect to VEH (animals administered with LPS 5 mg/kg). Values represent the mean ± SEM. The variables were evaluated by one-way analysis of variance (one-way-ANOVA) and the Dunnett post hoc test (****) *p* < 0.0001 vs. VEH group. No significant difference (ns).

**Figure 2 plants-12-02600-f002:**
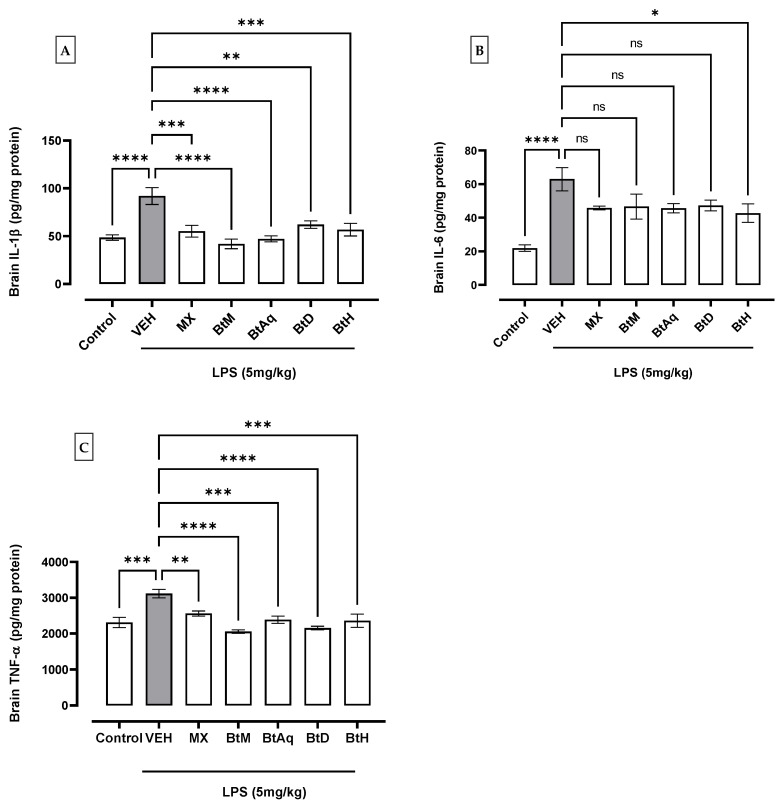
Effect of B. *ternifolia* extracts on the concentration of (**A**) IL-1β, (**B**) IL-6, and (**C**) TNF-α in the brain with respect to the VEH (animals administered with LPS (5 mg/kg). Values represent the mean ± SEM. The variables were evaluated by one-way analysis of variance (one-way-ANOVA) and the Dunnett post hoc test. (*) *p* < 0.05 (**) *p* < 0.01, (***) *p* < 0.001, (****) *p* < 0.0001 vs. VEH group. No significant difference (ns).

**Figure 3 plants-12-02600-f003:**
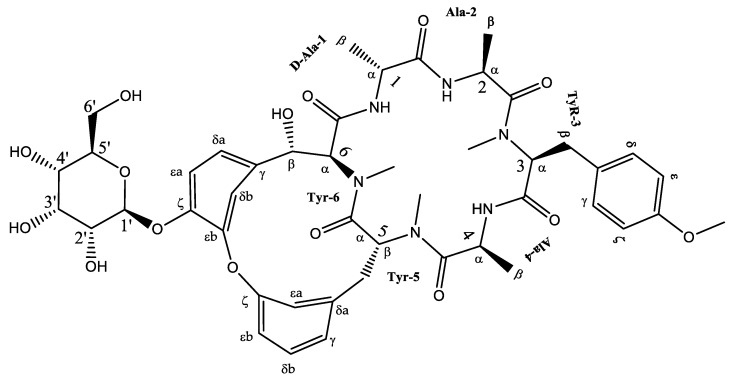
Chemical structure of the cyclic hexapeptide isolated from *B. ternifolia* root.

**Figure 4 plants-12-02600-f004:**
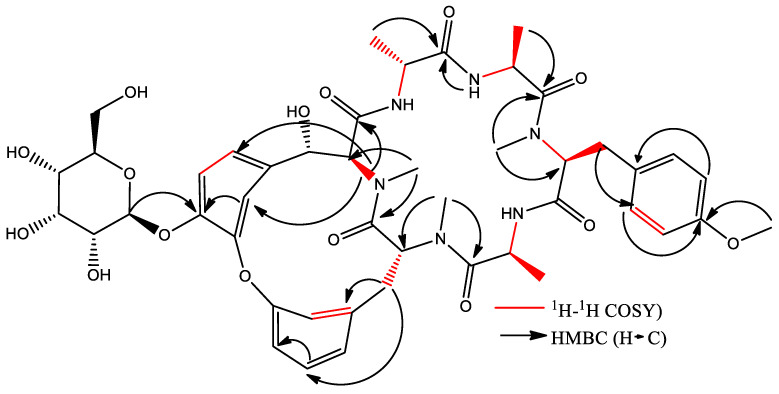
^1^H-^1^H COSY and HMBC correlations of the cyclic hexapeptide of *B. ternifolia*.

**Figure 5 plants-12-02600-f005:**
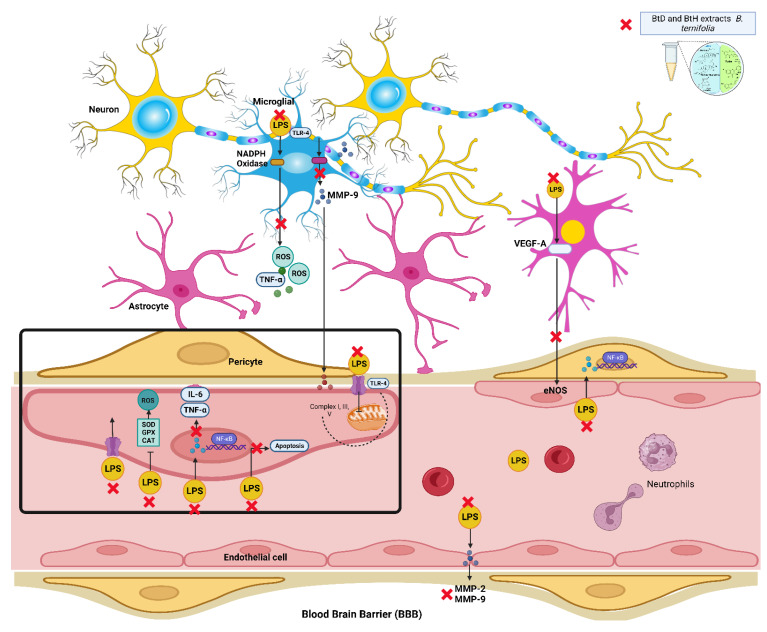
The schematic representation illustrates the anti-inflammatory and neuroprotective properties of *B. ternifolia* root extracts on the blood-brain barrier. Created with Biorender.com.

**Table 1 plants-12-02600-t001:** GC–MS analysis of dichloromethane extract of *B. ternifolia*.

No	RT (min)	Chemical Structure	Compound Name	Formula	MolecularWeight(g/mol)	Area(%)	Spectral InformationID
1	6.8	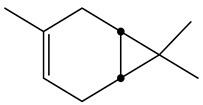	(M) 3-Carene	C_10_H_16_	136	1.2	103751
2	19.8	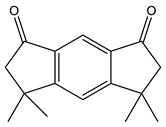	(M) s-indacene-1,7-dione,2,3,5,6-tetrahydro-3,3,5,5-tetramethyl	C_16_H_18_O_2_	242	14.3	31005
3	20.2	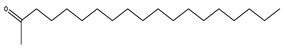	(M) 2-Nonadecanone	C_19_H_38_O	282	2.9	114582
4	21.1	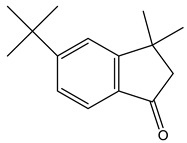	(M) 1H-Inden-1 one,5-(1,1-dimethylethyl)-2,3-dihydro-3,3-dimethyl	C_15_H_20_O	216	11.3	JP005240
5	28.8	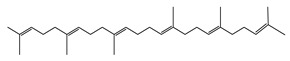	(R) Squalene	C_30_H_50_	410	9.13	2140
6	32.7	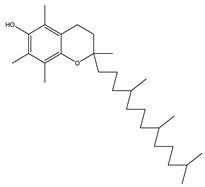	(M) D, α-Tocopherol	C_29_H_50_O_2_	430	35.6	30334
7	34.7	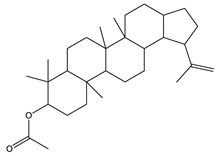	(M) Lup-20(29)-en-3-ol-acetate, (3β)	C_32_H_52_O_2_	468	5.4	194307

**Table 2 plants-12-02600-t002:** GC–MS analysis of hexane extract of *B. ternifolia*.

No	RT (min)	Chemical Structure	Compound Name	Formula	MolecularWeight(g/mol)	Area(%)	Spectral InformationID
8	18.4	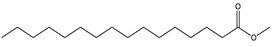	(R) Hexadecanoic acid, methyl ester	C_17_H_34_O_2_	270	3.2	1656
9	19.1	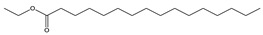	Hexadecanoic acid, ethyl ester	C_18_H_36_O_2_	284	0.8	27658
10	20.1	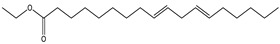	(M) 9,12-octadecadienoic acid (z-z)-methyl ester	C_19_H_34_O_2_	294	3.4	112-63-0
11	20.1	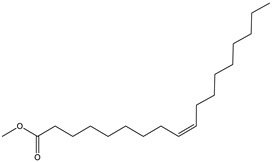	(M) 9- octadecadienoic acid (z)-methyl ester	C_19_H_36_O	296	2.4	JP005355
12	20.3	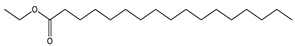	(R) Octadecanoic acid, methyl ester	C_19_H_38_O_2_	298	0.6	1815
13	20.7	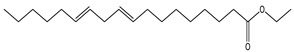	(M) 9,12 Octadecanoic acid, ethyl ester	C_20_H_36_O_2_	308	1.2	249157
14	20.7	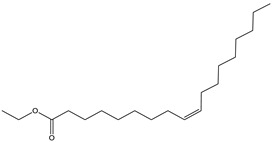	(R) Ethyl oleate	C_20_H_38_O_2_	310	1.4	JP002528
15	22.4	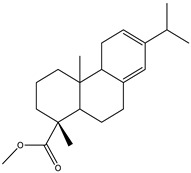	(M) Phenanthrenecarboxilic acid 1,2,3,4,4a,9,10,10a-dimethyl-7-(1-methylethyl)-methyl ester, [1R-(1α,4aβ,10aα)]	C_20_H_26_O_3_	316	3.8	3513-69-7
16	22.9	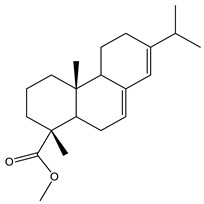	(R) methyl abietate	C_21_H_32_O_2_	316	0.6	JP003373
17	28.8	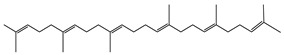	(R) Scualene	C_30_H_50_	410	20	2140
18	32.6	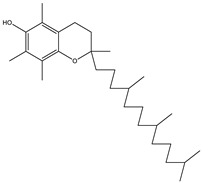	(M) D, α-Tocopherol	C_29_H_50_O_2_	430	44.8	30334
19	35.6	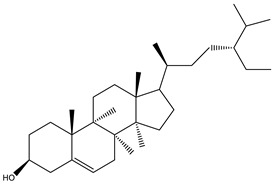	(R) β-Sitosterol	C_29_H_50_O	414	4.8	2300
20	44.7	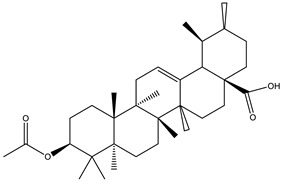	(T) Urs-12-en-28-al,3-(acetyloxy)-(3β)	C_32_H_50_O_3_	482	6.2	86996-88-5

**Table 3 plants-12-02600-t003:** 1H-NMR and 13C-NMR spectroscopy data of the cyclic hexapeptide *B. ternifolia* in (CD_3_OD δ in ppm, *J* in Hz).

Position		1H	13C
D-Ala1	α	4.48 (overlap)	48.3 d
	β	1.23 (overlap)	19.9 q
	C=O		172.4 s
Ala2	α	4.71 (overlap)	46.6 d
	β	1.28 (overlap)	17.5 q
	C=O		174.8 s
Tyr3	α	3.76 (overlap)	67.4 d
	βa	3.27 (m)	32.4 t
	γ		130.7 s
	δ*2	7.10 (8.5)	130.2 d
	ε*2	6.88 (8.7)	114.7 d
	ζ		159.3 s
	C=O		172 s
	NMe	2.96 (s)	38.8 q
	OMe	3.77 (s)	54.2 q
Ala4	α	4.71 (overlap)	47.02 d
	β	1.12 (d,6.6)	17.5 q
	C=O		172.2 s
Tyr5	α	5.46 (dd,11.6,3.1)	57.5 t
	βa	2.63 (d,11.2)	38.9
	βb	3.57 (overlap)	
	γ		139.3 s
	δa	7.13 (overlap)	130.9 d
	δb	7.52 (dd,8.6,2.0)	130.2 d
	εa	7.31 (dd,8.3,2.3)	123.9 d
	εb	7.42 (dd,8.5,2.2)	125.5 d
	ζ		159.3 s
	C=O		172.1 s
	NMe	3.2 (s)	29.2 q
Tyr6	α	4.55 (overlap)	67.4 d
	βa	4.48 (d,6.9)	73.5 d
	βb		
	γ		135.8 s
	δa	6.95 (dd,8.6,2.4)	125.5 d
	δb	5.0 (d,7.8)	117.4 d
	εa	7.10 (d,8.5)	121.3 d
	eb		153.1 s
	ζ		144.2 s
	C=O		170.5 s
	NMe	2.3 (s)	32.5 s
Glucose	1′	5.0 (d,7.8)	101.5 d
	2′	3.56 (overlap)	76.5 d
	3′	3.48 (m)	76.9 d
	4′	3.44 (overlap)	70.0 d
	5′	3.44 (overlap)	77.6 d
	6′	3.77 (dd,12.0, 4.8)	61.1 t
		3.87 (overlap)	

a,b = hydrogen position; s = primary carbon; d = secondary carbon; t = tertiary carbon; q = quaternary carbon.

**Table 4 plants-12-02600-t004:** Groups and experimental treatments with LPS (5 mg/kg).

Group	Oral Treatment	Doses
Control	Water	100 µL/10 g
VEH	Water + LPS *	5 mg/kg *
Experimental	MX BtM BtAq BtD BtH	25 mg/kg for 5 days

* LPS was administered 4 h prior to injecting the mouse with Evans blue. MX = meloxicam; BtM = methanolic extract; BtAq = aqueous extract; BtD = dichloromethane extract; BtH = hexanic extract.

## Data Availability

Not applicable.

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
