# Peer review of "Cyclic Hexapeptide from Bouvardia ternifolia (Cav.) Schltdl. and Neuroprotective Effects of Root Extracts"

_plants, 2023, doi:10.3390/plants12142600_

Round 1

Reviewer 1 Report

The manuscript presents results on the composition of different root extracts of plant shrub Bouvardia ternifolia obtained by GC-MS analysis and the effect of some of these extracts on protecting the blood-brain barrier. The work is interesting but the way of presentation seems a bit confusing. The aim of the study and the methodology should be more clearly presented. only in the conclusion do the authors more clearly present the sequence of conducting research and making decisions.

The Abstract should be more concise and clearly present the aim of the paper. The parts of the sentences in line 15 “…, the identification of compounds” as well as in the line 17 “…, and the cyclic heptapeptide….” are unclear and should be rewritten.

In presenting the results, authors begin with the detected cyclic heptapeptide, and the title itself brings it into focus, but I fail to see the higher importance of this finding in regard to other results in this paper. Especially because it is stated in the discussion that this compound has already been found in root extracts from Rubia yunnanensis.  I suggest to rearrange the results in order to obtain a more meaningful sequence of the research itself.

In the discussion,

Line285, the sentence “As with any medication…” the sentence is unclear

everything from the line 295 is presenting how and why was the experiment on the permeability of BBB but the mention and the discussion on their own results is missing.

There is a lot of Supplementary material provided, but I suggest to reorder their sequence as mentioned in text. Also, S2-S4 are not mentioned in text at all. And, line 192 S9 is actually S10, and in line 171 S10 is S11

Line 176 the paragraph 2.5. is missing

English language should be checked, some sentences are unclear or missing a verb.

English language should be checked, some sentences are unclear or missing a verb.

Author Response

Manuscript

ID plants-2464785

Response to Reviewer 1 Comments

The manuscript presents results on the composition of different root extracts of plant shrub Bouvardia ternifolia obtained by GC-MS analysis and the effect of some of these extracts on protecting the blood-brain barrier. The work is interesting but the way of presentation seems a bit confusing.

  1. The aim of the study and the methodology should be more clearly presented. only in the conclusion do the authors more clearly present the sequence of conducting research and making decisions.

Response: We appreciate your comments. The aim of the study was rewritten, and methodology was revised. We hope that now is clear.

  1. The Abstract should be more concise and clearly present the aim of the paper. The parts of the sentences in line 15 “…,the identification of compounds” as well as in the line 17 “…,and the cyclic heptapeptide….” are unclear and should be rewritten

Response: The abstract was revised; we hope that now is clear.

  1. In presenting the results, authors begin with the detected cyclic heptapeptide, and the title itself brings it into focus, but I fail to see the higher importance of this finding in regard to other results in this paper. Especially because it is stated in the discussion that this compound has already been found in root extracts from Rubia yunnanensis. I suggest to rearrange the results in order to obtain a more meaningful sequence of the research itself.

Response: The results section was rearranged as suggested.

  1. Line285, the sentence “As with any medication…” the sentence is unclear everything from the line 295 is presenting how and why was the experiment on the permeability of BBB but the mention and the discussion on their own results is missing.

Response: Discussion was revised and rearranged based on the presentation of the results.

  1. There is a lot of Supplementary material provided, but I suggest to reorder their sequence as mentioned in text. Also, S2-S4 are not mentioned in text at all. And, line 192 S9 is actually S10, and in line 171 S10 is S11 Line 176 the paragraph 2.5. is missing

Response: Supplementary material section was revised, and now it contains the most significant data, and in a logical sequence.

  1. English language should be checked, some sentences are unclear or missing a verb.

Response: The text was checked.

Reviewer 2 Report

The article “Cyclic hexapeptide from Bouvardia ternifolia  (Cav.) Schltdl. and neuroprotective effects of root extracts” represents novel and scientifically relevant data on an insufficiently researched plant species Bouvardia ternifolia.

The research methodology and concept of the manuscript are relevant and provide useful scientific data on bioactive compounds from Bouvardia ternifolia root and their anti-inflammatory and neuroprotective potential.

However, there are some disadvantages of this article that should be improved.

The authors emphasised that methanolic and aqueous extracts were not subjected to chemical analysis (abstract, line 19-20). It should be explained why dichloromethane and hexane extracts were only analysed.

The main problem in the methodology is related to GC-MS analysis and the identification of the compounds. The authors stated that identification was based on the comparison of the obtained spectra with the NIST library. It is mandatory to have at least one confirmation of the results (especially in the case of not so common compounds), from the literature or by calculating retention indexes. Please provide this data.

There is also inconsistency in the presentation of the results (Tables 2 and 3). In Table 2 the first compound is the one with the highest retention time while in Table 3 it is the other way around.

The discussion part has certain inconsistencies.

Lines 256-259 should be incorporated with the part about BBB crossing (from line 295) or removed from the text.

I suggest that the part of the discussion related to meloxicam (lines 283-294) be shortened. I can see the point of this part but since extracts were not tested for potential side effects it is not necessary to point out the side effects of the control medication.

The discussion about BBB is too general and too comprehensive without clear conclusions about the mechanism of action of the extracts (315-334). This part should be shortened, and potential mechanisms of similar plant extracts (tested od BBB) should be incorporated in the discussion.

In the whole discussion part, there is no explanation of how cytokines are relevant for neuroprotection. Instead of discussing single components and their impact on cytokines (or at least discuss it on a smaller scale), it would be more appropriate to discuss potential mechanisms of neuroprotection and the impact of natural products on cytokines.

The conclusions should be better elaborated.

Please uniform citation style in the Reference part considering the guidelines of the journal.

There are some minor typing mistakes.

Author Response

Manuscript

ID plants-2464785

Response to Reviewer 2 Comments

  1.  The article “Cyclic hexapeptide from Bouvardia ternifolia (Cav.) Schltdl. and neuroprotective effects of root extracts” represents novel and scientifically relevant data on an insufficiently researched plant species Bouvardia ternifolia. The research methodology and concept of the manuscript are relevant and provide useful scientific data on bioactive compounds from Bouvardia ternifolia root and their antiinflammatory and neuroprotective potential. However, there are some disadvantages of this article that should be improved.

Response: We appreciate your comments.

  1.  The authors emphasised that methanolic and aqueous extracts were not subjected to chemical analysis (abstract, line 19-20). It should be explained why dichloromethane and hexane extracts were only analysed.

Response: Dichloromethane and hexane extracts were the two extracts with significant activity. So, were the only extracts chemically characterized. Reviewer 1 suggested that the results section was rearranged, so now, we first present results of blood-barrier assay ant then chemical characterization. We hope now it is clear.

  1. The main problem in the methodology is related to GC-MS analysis and the identification of the compounds. The authors stated that identification was based on the comparison of the obtained spectra with the NIST library. It is mandatory to have at least one confirmation of the results (especially in the case of not so common compounds), from the literature or by calculating retention indexes. Please provide this data.

Response: The data for the calculation of retention rates were not carried out. the results were confirmed through what was reported in the literature comparing the TIC chromatogram, the mass and the fragmentation patterns, the ID of the spectrum information was added in tables 1 and 2.

  1.  There is also inconsistency in the presentation of the results (Tables 2 and 3). In Table 2 the first compound is the one with the highest retention time while in Table 3 it is the other way around.

Response: Tables 1 and 2 were reviewed and the order of retention times was modified.

  1. The discussion part has certain inconsistencies. Lines 256-259 should be incorporated with the part about BBB crossing (from line 295) or removed from the text.

Response: lines 256-259 were removed from the text

  1. I suggest that the part of the discussion related to meloxicam (lines 283-294) be shortened. I can see the point of this part but since extracts were not tested for potential side effects it is not necessary to point out the side effects of the control medication.

Response: The discussion related to meloxicam was shortened as suggested and focused on its mechanism of action.

  1. The discussion about BBB is too general and too comprehensive without clear conclusions about the mechanism of action of the extracts (315-334). This part should be shortened, and potential mechanisms of similar plant extracts (tested od BBB) should be incorporated in the discussion.

Response: Lines 315-334 were shortened as suggested, and discussion of potential mechanisms of plant extracts tested on BBB is now included.

  1. In the whole discussion part, there is no explanation of how cytokines are relevant for neuroprotection. Instead of discussing single components and their impact on cytokines (or at least discuss it on a smaller scale), it would be more appropriate to discuss potential mechanisms of neuroprotection and the impact of natural products on cytokines.

Response: Discussion was modified as suggested, and potential mechanisms of neuroprotection and the impact of natural products on cytokines are now included.

  1. Please uniform citation style in the Reference part considering the guidelines of the journal.

Response: The reference part was revised, and now it is presented considering the guidelines of the journal

  1.  There are some minor typing mistakes.

Response: The manuscript was checked for typing mistakes.

Round 2

Reviewer 1 Report

The authors have addresed all comments from the previous review and improved the overall meritt of the paper and therefore I suggest the manuscript to be accepted for publications

Reviewer 2 Report

Before the publication of the manuscript please adjust the reference style according to the instructions which can be found on Plants journal webpage:

References should be described as follows, depending on the type of work:

  • Journal Articles:
    1. Author 1, A.B.; Author 2, C.D. Title of the article. Abbreviated Journal Name Year, Volume, page range.